# Towards Adaptive Video Stabilization: A Robustness Benchmark of IMU Predictors for Cascaded Routing

## Abstract

Deploying energy-intensive computer vision algorithms on edge devices requires architectural shifts to minimize computational overhead. We propose a **cascaded inference** paradigm for adaptive video stabilization, utilizing a lightweight Inertial Measurement Unit (IMU)-based model to route motion anomalies to heavy verification solvers. To establish a rigorous foundation for this **cascaded router**, we benchmark 12 multi-paradigm models aligned with **Trustworthy AI** principles: evaluating predictive accuracy, strict single-sample sequential latency, and out-of-distribution (OOD) robustness under severe sensor noise. Additionally, **Game-Theoretic SHAP analysis** extracts physical kinematics, quantifying translational (accelerometer) versus rotational (gyroscope) contributions to inform downstream stabilization strategies (OIS vs. EIS). Our results reveal that standard batch-latency metrics mask significant API overheads in gradient boosting frameworks. Consequently, **CatBoost** emerges as the optimal deterministic router ($0.44$ ms), minimizing serialization costs. Crucially, our evaluation of the probabilistic **Bayesian 1D-CNN** exposes a vulnerability to *confident misclassification* under severe noise, demonstrating that epistemic uncertainty alone is insufficient for fail-safe routing. However, achieving sub-millisecond execution ($0.38$ ms) and near-perfect accuracy, the Bayesian architecture provides mathematically rigorous epistemic bounds. This formally justifies orthogonal Isolation Forest integration for OOD interception. Ultimately, by coupling high-speed predictive routing, mathematically rigorous safety bounds, and game-theoretic XAI, this work advances the formal basis for designing reliable hybrid edge-AI systems.

**Keywords:** Trustworthy AI, Explainable AI, Shapley Values, Bayesian Neural Networks, Edge Computing, Cascaded Inference, Anomaly Detection.

## 1 Introduction

The ubiquity of high-resolution video recording on edge devices, such as smartphones and wearable action cameras, poses computational challenges. Modern devices routinely capture video at 4K resolution and 60 frames per second. However, raw handheld footage is susceptible to high-frequency mechanical vibrations and low-frequency motion anomalies (e.g., walking, running, or riding a scooter). To mitigate this, contemporary video stabilization relies on Optical Image Stabilization (OIS), Electronic Image Stabilization (EIS), and deep learning-based post-processing methods like dense optical flow or 3D perspective rendering. While these vision-based algorithms deliver state-of-the-art smoothness, they impose a severe computational bottleneck. Executing continuous, per-pixel dense neural networks on every high-resolution frame results in thermal throttling, battery depletion, and unacceptable latency on edge devices. Moreover, motion anomalies are not uniformly distributed across time; periods of severe shaking are often interspersed with relative stability. Consequently, processing every frame with the same parameter-heavy visual solver is inefficient. This highlights the need for a paradigm shift towards **conditional computation** and **adaptive routing**.

To address this, we propose an Inertial Measurement Unit (IMU)-based early-stage predictive model embedded within a **Cascaded Inference** architecture. IMU sensors (accelerometers and gyroscopes) operate at high frequencies (e.g., 100–200 Hz) but consume orders of magnitude less power than im-

age signal processors (ISPs). By monitoring low-dimensional kinematic data, a lightweight "Smart Router" formulates hypotheses regarding motion anomalies. If the kinematic stream indicates stability, the heavy downstream video solver is bypassed. Conversely, if a severe shake is detected, the router triggers energy-intensive vision algorithms. However, deploying such a router on mobile hardware introduces stringent constraints. The predictor must operate with sub-millisecond sequential latency to prevent buffer overruns. More importantly, it must adhere to **Trustworthy Artificial Intelligence** principles. In real-world edge scenarios, IMU sensors face severe mechanical noise, thermal drift, and out-of-distribution (OOD) shocks. Standard deterministic models (e.g., uncalibrated deep networks or decision trees) tend to fail silently under noise, outputting overconfident but incorrect predictions that could misguide the downstream solver. Therefore, building a reliable router requires mathematically sound uncertainty quantification and physical explainability.

In this paper, we establish the mathematical and empirical foundation for such an adaptive stabilization pipeline. We present a robustness benchmark of 12 multi-paradigm predictive models, tailored for edge-based kinematic routing. The primary **contributions** of this work are threefold:

- **Comprehensive Robustness Benchmarking for Edge Inference:** We evaluate 12 algorithms across four mathematical paradigms (Linear/Kernel methods, Tree Ensembles, Deep Neural Networks, and Probabilistic Architectures). Crucially, we eschew standard amortized batch-latency metrics in favor of a strict **single-sample sequential latency test**. This exposes hidden serialization overheads in popular frameworks, ensuring hardware realism. Models are stress-tested against OOD anomalies and severe Gaussian noise ($\sigma = 0.8$) to quantify structural degradation.

- **Game-Theoretic Physical Mapping via SHAP:** Explainability serves as an active routing mechanism. We apply **Shapley Additive exPlanations (SHAP)**, rooted in cooperative game theory, to mathematically quantify marginal contributions of sensor axes. By dynamically resolving whether an anomaly is translational (accelerometer-driven) or rotational (gyroscope-driven), the predictor instructs the video solver to prioritize perspective cropping (OIS) or angular transformation (EIS).

- **Exposing the Limits of Epistemic Uncertainty:** We integrate a **Bayesian 1D-CNN** using variational inference to evaluate uncertainty-guided routing. Stress-tests reveal a counterintuitive *confident misclassification* phenomenon under severe noise, proving Bayesian bounds alone cannot guarantee safety on low-dimensional kinematics. This justifies our hybrid topological design, combining Isolation Forests for OOD interception with high-speed **predictive routers**. *To support reproducibility, all code and benchmark configurations will be publicly released upon acceptance (currently anonymized for double-blind review).*

By bridging hardware-efficient machine learning with probabilistic safety bounds and game-theoretic explainability, this work advances trustworthy hybrid AI systems for mobile vision.

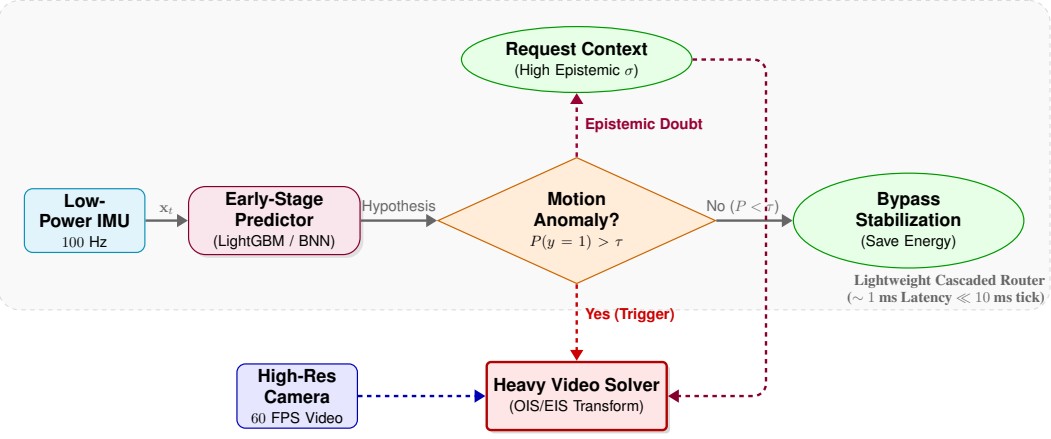

Figure 1: Proposed Cascaded Inference Architecture. The lightweight IMU predictor evaluates motion kinematics. The energy-intensive *Heavy Video Solver* is triggered when anomalous shaking is detected ($P > \tau$) or the Bayesian predictor flags high epistemic uncertainty ($\sigma_{epistemic} > \epsilon$).

## 2 RELATED WORK

Our proposed architecture lies at the intersection of cross-modal sensor fusion, conditional edge computing, and trustworthy artificial intelligence.

**Deep Video Stabilization and Sensor Fusion.** Modern video stabilization has transitioned from classical feature matching to deep learning paradigms, utilizing dense optical flow (Teed & Deng, 2020) and 3D perspective warping (Liu et al., 2021). To handle extreme motion, recent approaches have increasingly integrated IMU data (gyroscope and accelerometer) to guide the visual warping field (Ouyang et al., 2022; Zhao et al., 2023). However, these state-of-the-art methods operate under a *tightly-coupled* fusion paradigm, where both the high-frequency IMU stream and the heavy high-resolution video frames are processed simultaneously at every temporal step. While this maximizes visual smoothness, it inherently fails to address the computational bottleneck on edge devices. Our work diverges by proposing a *loosely-coupled*, cascaded architecture where the IMU acts strictly as a low-power gatekeeper, bypassing the visual solver entirely during stable kinematics.

**Cascaded Inference and Conditional Computation on the Edge.** To mitigate the latency and energy constraints of mobile devices, dynamic routing and conditional computation have gained significant traction. Approaches such as early-exit neural networks (Han et al., 2021) and cascaded classifiers (Boluk et al., 2021) allow "easy" inputs to be processed by shallow sub-networks, reserving deeper layers for complex data. More recently, speculative decoding has emerged to accelerate autoregressive generation by drafting hypotheses with lightweight models (Chen et al., 2023). While lightweight architectural designs have also been successfully deployed for efficient, continuous IMU kinematics processing (Wang et al., 2026), most existing dynamic routing frameworks operate within a single modality (e.g., exclusively IMU or exclusively video) and frequently overlook the stark hardware reality of API serialization overheads in sequential single-sample processing. We extend the speculative routing paradigm to cross-modal edge processing (IMU to Video) and expose the critical latency penalties of standard boosting frameworks in unbatched scenarios.

**Trustworthy AI: Uncertainty and Game-Theoretic Explainability.** Deploying AI in physical edge environments requires resilience to out-of-distribution (OOD) mechanical shocks and sensor noise. While deterministic models often yield overconfident, miscalibrated predictions under domain shift (Gawlikowski et al., 2023), Bayesian Neural Networks (BNNs) offer a mathematically rigorous framework for quantifying epistemic uncertainty via variational inference (Jospin et al., 2022). Concurrently, Explainable AI (XAI) techniques, notably game-theoretic Shapley values (Lundberg & Lee, 2017), are standard for post-hoc model debugging. While recent literature strongly advocates for deploying actionable XAI in edge environments to ensure system reliability (Wang et al., 2024), our framework uniquely synthesizes these pillars. We elevate SHAP from a visualization tool to a functional kinematics decoder, utilizing the marginal contributions of sensor axes to dynamically select the optimal downstream compensation strategy (OIS vs. EIS).

## 3 MATHEMATICAL FRAMEWORK OF CASCADED INFERENCE

The core principle of our architecture is decoupling high-frequency kinematic anomaly detection from low-frequency, resource-intensive visual processing. Figure 1 illustrates this workflow. In this section, we formally define the mathematical mechanisms of the early-stage predictor, focusing on conditional routing, anomaly detection, and robustness evaluation.

### 3.1 FORMALIZATION OF THE CASCADED ROUTER AND UNCERTAINTY ROUTING

Let $\boldsymbol{x}_t \in \mathbb{R}^d$ denote a continuous multi-dimensional time-series vector from the device's IMU (accelerometer and gyroscope readings) at timestamp $t$, and let $\boldsymbol{I}_t$ represent the corresponding high-resolution camera frame. The goal of adaptive stabilization is to execute a heavy, non-linear video transformation function $f_{\text{video}}(\boldsymbol{I}_t)$ (e.g., dense optical flow or perspective cropping) *only* when necessary. To achieve this, we introduce an early-stage predictive model $g_{\boldsymbol{\theta}}(\boldsymbol{x}_t)$, parameterized by weights $\boldsymbol{\theta}$. In a deterministic setting, this model maps kinematic input to a discrete probability distribution over binary classes (Stable vs. Motion Anomaly):

$$p_t = P(\text{y} = 1 \mid \boldsymbol{x}_t; \boldsymbol{\theta}) \tag{1}$$

where y $= 1$ indicates severe shaking requiring stabilization.

To ensure safe routing under domain shift, we expand $g_{\boldsymbol{\theta}}$ to a probabilistic framework (e.g., via a Bayesian Neural Network). In this paradigm, the model outputs not only the expected probability $p_t$ but also quantifies its *epistemic uncertainty* $\sigma_{\text{epistemic}}(\boldsymbol{x}_t)$, which reflects the model's ignorance about the data generating process and increases when $\boldsymbol{x}_t$ lies in a sparsely populated region of the feature space. Consequently, we define the activation state $a_t \in \{0, 1\}$ of the downstream heavy video solver via the indicator function $\mathbb{I}(\cdot)$:

$$a_t = \mathbb{I}\Big((p_t > \tau) \vee (\sigma_{\text{epistemic}}(\boldsymbol{x}_t) > \epsilon)\Big) \tag{2}$$

where $\tau \in [0, 1]$ is the anomaly probability threshold, and $\epsilon > 0$ is the maximally tolerated epistemic uncertainty boundary. Equation 2 establishes a **fail-safe routing protocol**: the video solver is invoked when a shake is confidently detected ($p_t > \tau$) or when the router lacks confidence to bypass it ($\sigma_{\text{epistemic}} > \epsilon$). If $a_t = 0$, the frame bypasses stabilization, conserving resources.

## 3.2 TRUSTWORTHY AI: OUT-OF-DISTRIBUTION (OOD) SAFETY MONITORING

While epistemic uncertainty captures model doubt, we incorporate an orthogonal safety mechanism to detect hardware faults, sensor disconnections, or novel physical environments—situations categorized as Out-of-Distribution (OOD) data. To address this, we utilize the Isolation Forest algorithm, which isolates anomalies rather than profiling normal points. For a kinematic sample $\boldsymbol{x}$, the Isolation Forest constructs an ensemble of random trees and calculates the average path length $\mathbb{E}[h(\boldsymbol{x})]$ required to isolate the sample. The anomaly score $s(\boldsymbol{x})$ is defined as:

$$s(\boldsymbol{x}) = 2^{-\frac{\mathbb{E}[h(\boldsymbol{x})]}{c(n)}} \tag{3}$$

where $c(n)$ is the average path length of an unsuccessful search in a Binary Search Tree built from $n$ samples. If $s(\boldsymbol{x})$ approaches 1, the sample is highly anomalous (OOD). Specifically, if $s(\boldsymbol{x}) > \gamma$, the router halts inference and flags a hardware-level *fail-safe* event, protecting the downstream solver from corrupted sensor streams.

## 3.3 MATHEMATICAL FORMULATION OF THE ROBUSTNESS TEST

Edge devices face severe physical shocks, electromagnetic interference, and thermal degradation, compromising the signal-to-noise ratio (SNR) of IMU sensors; therefore, a trustworthy router must resist catastrophic performance degradation under such conditions. To benchmark stability, we define a noisy observation model where the observed sensor reading $\tilde{\boldsymbol{x}}_t$ is corrupted by additive white Gaussian noise (AWGN):

$$\tilde{\mathbf{x}}_t = \mathbf{x}_t + \boldsymbol{\varepsilon}, \quad \boldsymbol{\varepsilon} \sim \mathcal{N}(\mathbf{0}, \sigma^2 \mathbf{I}) \tag{4}$$

where $\boldsymbol{I}$ is the identity matrix and $\sigma^2$ is the noise variance. In our evaluation (Section 6), we simulate severe physical disruption by setting $\sigma = 0.8$ (applied to standardized features). We evaluate the structural robustness of $g_{\boldsymbol{\theta}}$ by calculating the predictive degradation metric $\Delta_{\text{Drop}}$:

$$\Delta_{\text{Drop}} = \text{Accuracy}\big(g_{\boldsymbol{\theta}}(\boldsymbol{x})\big) - \text{Accuracy}\big(g_{\boldsymbol{\theta}}(\tilde{\boldsymbol{x}})\big) \tag{5}$$

Models with lower $\Delta_{\text{Drop}}$ are prioritized for deployment, as they demonstrate invariant decision boundaries robust to high-entropy sensor perturbations.

## 4 TAXONOMY OF EVALUATED PREDICTIVE MODELS

To identify the optimal framework for the kinematic predictor $g_{\boldsymbol{\theta}}(\boldsymbol{x})$, we benchmarked 12 machine learning architectures. Figure 2 shows models categorized into four paradigms. This section describes these groups, emphasizing two promising architectures for cascaded pipeline: **LightGBM** (for extreme inference speed) and **Bayesian 1D-CNN** (for epistemic uncertainty quantification).

### 4.1 LINEAR AND KERNEL METHODS

This group relies on linear separability or implicit mappings into Hilbert spaces. Logistic Regression serves as the baseline, modeling the posterior probability as $P(\text{y} = 1|\boldsymbol{x}) = \sigma(\boldsymbol{w}^\top \boldsymbol{x} + b)$. To capture non-linear patterns without the cost of exact kernel SVMs, we include **Random Fourier**

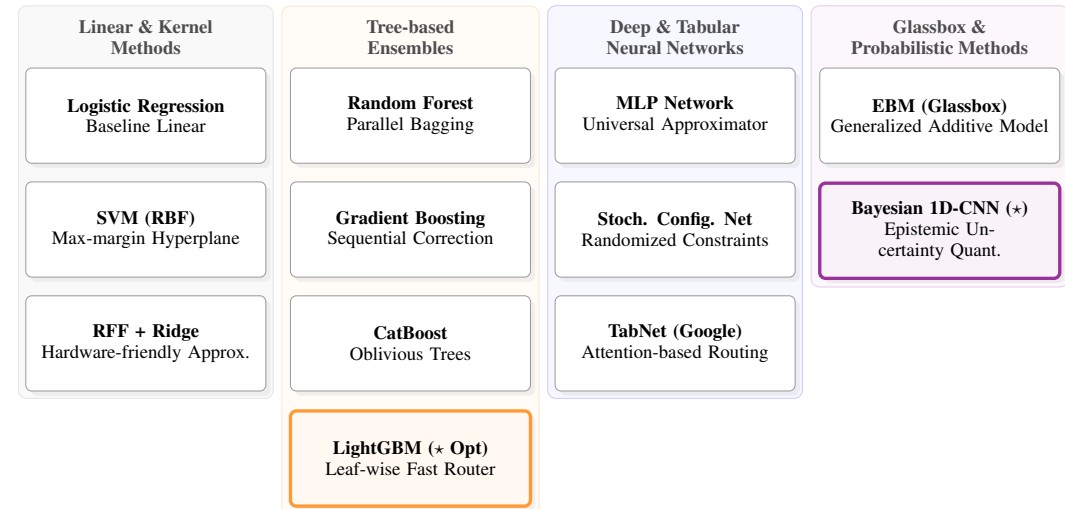

Figure 2: Taxonomy of the 12 mathematical frameworks evaluated for early-stage kinematic prediction. Models are functionally clustered into four distinct paradigms. **LightGBM** is highlighted as the optimal high-speed router, while the **Bayesian 1D-CNN** is emphasized for its unique capacity to provide mathematically rigorous epistemic uncertainty bounds via variational inference.

**Features (RFF)** coupled with Ridge Regression. RFF approximates the shift-invariant RBF kernel $k(\boldsymbol{x}, \boldsymbol{y}) \approx \boldsymbol{z}(\boldsymbol{x})^\top \boldsymbol{z}(\boldsymbol{y})$ using a randomized feature map:

$$\boldsymbol{z}(\boldsymbol{x}) = \sqrt{\frac{2}{D}} \cos(\boldsymbol{W}\boldsymbol{x} + \boldsymbol{b}) \tag{6}$$

where $\boldsymbol{W} \sim \mathcal{N}(\boldsymbol{0}, \gamma\boldsymbol{I})$ and $\boldsymbol{b} \sim \text{Uniform}(0, 2\pi)$. This reduces infinite-dimensional kernel computation to hardware-friendly matrix multiplication, suitable for edge microcontrollers.

## 4.2 Deep and Tabular Neural Networks

While MLPs act as universal approximators, their dense matrix multiplications yield suboptimal latency-to-accuracy ratios on tabular data (e.g., 9-axis IMU readings). To address this, we evaluate **Stochastic Configuration Networks (SCN)**, which incrementally build hidden layers subject to inequality constraints, and **TabNet**, proposed by Google, employing sequential attention. TabNet utilizes sparse instance-wise feature selection masks $\boldsymbol{M}[i] = \text{sparsemax}(\boldsymbol{W}_a \boldsymbol{h}[i-1])$, learning to route kinematic signals through multi-step decisions.

## 4.3 Tree-Based Ensembles and LightGBM Optimization

GBDTs build an additive model $F_M(\boldsymbol{x}) = \sum_{m=1}^{M} \gamma_m h_m(\boldsymbol{x})$ by minimizing a regularized objective. We benchmark standard Gradient Boosting, Random Forest, CatBoost (utilizing oblivious symmetric trees), and **LightGBM**.

LightGBM suits sub-millisecond routing due to its **leaf-wise (best-first) tree growth** strategy. Unlike level-wise growth, LightGBM splits the leaf with maximum variance gain, regardless of depth. Let $\mathcal{D} = \{(\boldsymbol{x}_i, \mathrm{y}_i)\}_{i=1}^{n}$ be the training set, and let $g_i$ and $h_i$ denote the first and second-order gradients of the loss function. When splitting a node, the structural gain $\Delta\mathcal{L}$ is computed as:

$$\Delta\mathcal{L} = \frac{1}{2}\left[\frac{\left(\sum_{i\in I_L} g_i\right)^2}{\sum_{i\in I_L} h_i + \lambda} + \frac{\left(\sum_{i\in I_R} g_i\right)^2}{\sum_{i\in I_R} h_i + \lambda} - \frac{\left(\sum_{i\in I} g_i\right)^2}{\sum_{i\in I} h_i + \lambda}\right] \tag{7}$$

LightGBM accelerates this via *Gradient-based One-Side Sampling* (GOSS), prioritizing instances with large gradients ($|g_i|$), yielding asymmetric, deep trees. While asymmetric trees suggest near-instantaneous execution, edge deployment requires evaluating API serialization overhead for sequential streams (see Section 6).

## 4.4 Probabilistic Methods and Bayesian 1D-CNN

Deterministic networks and GBDTs cannot express *"I do not know"*, often outputting uncalibrated estimates collapsing to extremes (0.99 or 0.01) under OOD noise. To satisfy the uncertainty constraint $\sigma_{\text{epistemic}} < \epsilon$ (Eq. 2), we integrate a **Bayesian 1D Convolutional Neural Network (BNN)**. A BNN places a prior distribution $p(\boldsymbol{w}) \sim \mathcal{N}(\boldsymbol{0}, \boldsymbol{I})$ over parameters and seeks to infer the posterior $p(\boldsymbol{w}|\mathcal{D})$. Since the posterior is intractable for 1D-CNNs, we employ **Variational Inference**, introducing a parameterized variational distribution $q_{\boldsymbol{\theta}}(\boldsymbol{w})$ to approximate the true posterior. The optimization objective minimizes the KL divergence $D_{\text{KL}}(q_{\boldsymbol{\theta}}(\boldsymbol{w}) \| p(\boldsymbol{w}|\mathcal{D}))$, which is equivalent to maximizing the **Evidence Lower Bound (ELBO)**:

$$\mathcal{L}_{\text{ELBO}}(\boldsymbol{\theta}) = \mathbb{E}_{q_{\boldsymbol{\theta}}(\mathbf{w})}[\log p(\mathcal{D}|\mathbf{w})] - D_{\text{KL}}(q_{\boldsymbol{\theta}}(\mathbf{w}) \| p(\mathbf{w})) \tag{8}$$

Here, the first term represents expected log-likelihood, while the second penalizes deviation from the prior. We optimize this objective using the reparameterization trick, expressing weights as $\mathbf{w} = \boldsymbol{\mu} + \boldsymbol{\sigma} \odot \boldsymbol{\varepsilon}$, where $\boldsymbol{\varepsilon} \sim \mathcal{N}(\boldsymbol{0}, \mathbf{I})$.

During inference, given $\boldsymbol{x}^*$, we estimate the predictive distribution by sampling $T$ Monte Carlo weights $\hat{\boldsymbol{w}}_t \sim q_{\boldsymbol{\theta}}(\boldsymbol{w})$ from the optimized variational posterior. The final stabilization hypothesis $p^*$ and its **epistemic uncertainty** $\sigma_{\text{epistemic}}$ are computed as the empirical mean and variance of stochastic forward passes:

$$p^* \approx \frac{1}{T} \sum_{t=1}^{T} f(\boldsymbol{x}^*; \hat{\boldsymbol{w}}_t) \tag{9}$$

$$\sigma_{\text{epistemic}}^2 \approx \frac{1}{T} \sum_{t=1}^{T} \left( f(\boldsymbol{x}^*; \hat{\boldsymbol{w}}_t) - p^* \right)^2 \tag{10}$$

This framework fulfills Trustworthy AI requirements; by routing to the Heavy Solver when $\sigma_{\text{epistemic}}^2$ spikes, the BNN guarantees stabilization is not blindly applied to noisy shocks.

# 5 Experimental Setup and Evaluation Metrics

To validate our cascaded routing paradigm, we constructed an experimental pipeline focused on edge deployment constraints.

## 5.1 Dataset and Preprocessing

The benchmark uses a Hand Tremor Dataset collected with an MPU9250 IMU (Pandya & Dutta, 2024). The sensor captures 9-dimensional kinematic features ($\boldsymbol{x}_t \in \mathbb{R}^9$): 3-axis acceleration $(a_x, a_y, a_z)$, 3-axis angular velocity via gyroscope $(g_x, g_y, g_z)$, and 3-axis magnetic field strength $(m_x, m_y, m_z)$. The dataset simulates severe motion anomalies (e.g., scooter vibrations), providing a binary target: normal trajectory ($y = 0$) versus anomalous shaking ($y = 1$). To prevent leakage, features were standardized ($\mu = 0, \sigma = 1$) using statistics from training folds during cross-validation.

## 5.2 Trustworthy Evaluation Metrics

Models were evaluated using 5-Fold Stratified Cross-Validation. Beyond standard metrics (Accuracy, F1, ROC-AUC), we emphasize metrics critical for **Trustworthy AI**:

- **Noise Robustness ($\Delta_{\text{Drop}}$):** We evaluate decision boundary integrity by injecting severe additive white Gaussian noise ($\sigma = 0.8$) into the test set. The metric tracks the absolute accuracy drop.
- **Safety Monitoring (OOD AUROC):** The AUROC generated by the Isolation Forest, quantifying the ability to isolate hardware faults and out-of-distribution shocks.

## 5.3 On the Measurement of Edge Inference Latency

Standard batch-processed benchmarks often report artificially low latencies (e.g., $< 0.005$ ms) due to vectorized SIMD parallelization across large test sets. However, in smartphone deployment,

the predictor processes IMU data sequentially (batch_size = 1). To ensure hardware realism, our pipeline isolates sequential inference latency. We evaluate the models using a 100-iteration single-sample loop with cache warm-up. This bounds overhead, ensuring the router operates within the 100–200 Hz IMU sampling budget ($\approx$ 5–10 ms per tick) without buffer overruns.

## 5.4 Empirical Limitations: Confident Misclassification Under Noise

A prevailing assumption is that Bayesian Neural Networks (BNNs) prevent silent failures by yielding high epistemic uncertainty ($\sigma_{\text{epistemic}}$) on out-of-distribution or corrupted data. To test this, we evaluated the Bayesian 1D-CNN on the test set corrupted with severe mechanical noise ($\sigma = 0.8$).

Figure 3 illustrates epistemic uncertainty under this shift. Contrary to expectations, we observe **Confident Misclassification**. The error rate paradoxically peaks at over $50\%$ in the lowest uncertainty quantile ($0-10\%$), while predictions with highest doubt ($90-100\%$) exhibit lower error rates. Furthermore, the mean uncertainty for incorrect predictions ($\sigma = 0.00803$) is actually lower than for correct predictions ($\sigma = 0.00828$).

Mathematically, severe AWGN on low-dimensional data does not merely push samples into unpopulated latent space. Instead, noise translates samples across the decision boundary into high-confidence regions of the opposing class.

This vulnerability is critical for edge-AI. It proves relying exclusively on Bayesian uncertainty (Eq. 2) is insufficient for fail-safe routing under shocks. This justifies our design: the orthogonal **Isolation Forest** monitor (Eq. 3) is required to intercept OOD anomalies *before* classification, preventing catastrophic misrouting.

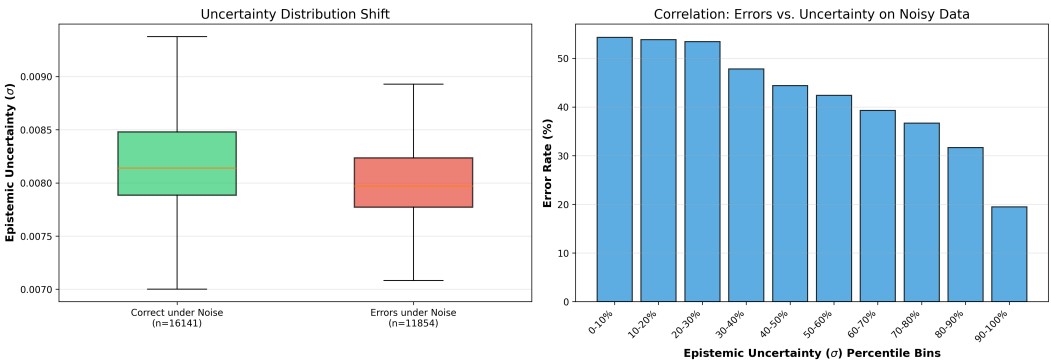

Figure 3: Epistemic Uncertainty Analysis of the Bayesian 1D-CNN under severe noise ($\sigma = 0.8$). The charts reveal *Confident Misclassification*, where highest error rates concentrate in lowest uncertainty bins. This justifies orthogonal OOD monitors like Isolation Forests in our pipeline.

## 6 Benchmark Results and Robustness Analysis

Table 1 summarizes performance of the 12 evaluated frameworks. Results reveal a trade-off between predictive accuracy, sequential latency, and noise robustness.

### 6.1 Decision Boundaries: Why Trees Fail Under Severe Noise

Table 1 reveals a disparity in structural robustness between linear methods and deep ensembles. Tree-based architectures (CatBoost, Random Forest, LightGBM) and neural models achieved near-perfect accuracy (F1 $\approx 0.997$). Under AWGN ($\sigma = 0.8$), these models suffered catastrophic degradation, dropping accuracy by $\approx 41\% - 45\%$. Logistic Regression and SCN maintained invariant boundaries, dropping only 11.7% and 7.5%.

This behavior stems from the topology of feature space partitioning. Tree ensembles use axis-aligned hyperplanes to partition latent space, creating sharp, localized boundaries fitted to training

data. High-variance noise easily perturbs samples across these boundaries. Linear and low-capacity kernel methods learn smooth, global, regularized hyperplanes; sacrificing capacity on clean data, their gradients absorb high-frequency perturbations.

## 6.2 Analysis of Sequential Latency and Probabilistic Efficiency

Our Single-Sample Edge Latency test reveals API serialization overhead in gradient boosting frameworks. While LightGBM excels in batch processing, its sequential latency (1.25 ms) lags CatBoost (0.44 ms) due to Python-to-C++ serialization overhead. Thus, CatBoost is the optimal *deterministic* router, balancing F1-score (0.9974) and sub-millisecond execution.

The **Bayesian 1D-CNN** offers key advantages for trustworthy edge deployment. At 0.38 ms latency, it surpasses the optimal boosting ensemble while quantifying epistemic uncertainty via variational inference. Unlike standard softmax probabilities that yield overconfident OOD predictions, this architecture provides calibrated confidence bounds. This enables the cascaded router to trigger the Heavy Solver based on statistical uncertainty, significantly enhancing the system's overall reliability in unpredictable environments.

Table 1: Benchmark Results for IMU Predictors. Metrics averaged across 5 folds. *Infer Time* reflects sequential single-sample latency (ms).

| MODEL | ACCURACY | AUC | F1-SCORE | ACC DROP (Noise) | OOD AUROC | INFER TIME (ms) |
|---|---|---|---|---|---|---|
| *Linear & Kernel Methods* | | | | | | |
| Logistic Regression | $0.7680 \pm 0.004$ | 0.8713 | 0.7681 | 0.1166 | 0.9826 | **0.0687** |
| SVM (RBF Kernel) | $0.9781 \pm 0.002$ | 0.9967 | 0.9781 | 0.4285 | 0.9828 | 0.2579 |
| RFF + Ridge | $0.9429 \pm 0.002$ | 0.9942 | 0.9430 | 0.3987 | 0.9827 | 1.2409 |
| *Tree-Based Ensembles* | | | | | | |
| Random Forest | $0.9972 \pm 0.000$ | 0.9998 | 0.9972 | 0.4241 | 0.9828 | 20.8740 |
| Gradient Boosting | $0.9958 \pm 0.001$ | 0.9999 | 0.9958 | 0.4275 | 0.9827 | 0.1786 |
| LightGBM | $0.9972 \pm 0.000$ | 1.0000 | 0.9972 | 0.4137 | 0.9829 | 1.2457 |
| CatBoost (**Opt. Det.**) | $\mathbf{0.9974 \pm 0.000}$ | **1.0000** | **0.9974** | 0.4205 | 0.9826 | 0.4439 |
| *Deep & Tabular Neural Networks* | | | | | | |
| MLP Neural Network | $0.9970 \pm 0.001$ | 0.9999 | 0.9970 | 0.4164 | 0.9830 | 0.1111 |
| SCN (Stochastic Config) | $0.6254 \pm 0.002$ | 0.9988 | 0.5425 | **0.0747** | 0.9829 | 2.8402 |
| TabNet (Google) | $0.9949 \pm 0.001$ | 0.9995 | 0.9949 | 0.4458 | 0.9828 | 4.2571 |
| *Probabilistic & Glassbox Methods* | | | | | | |
| EBM (Glassbox) | $0.9971 \pm 0.000$ | 0.9999 | 0.9971 | 0.4364 | 0.9827 | 0.1850 |
| **Bayesian 1D-CNN** | $0.9966 \pm 0.000$ | 0.9999 | 0.9966 | 0.4285 | 0.9827 | **0.3808** |

# 7 Game-Theoretic Kinematic Analysis via SHAP

While the Bayesian 1D-CNN provides rigorous bounds on epistemic uncertainty, designing a truly *Explainable AI* (XAI) system requires understanding the physical kinematics driving the anomaly detection. To achieve this without the prohibitive computational cost of approximating Shapley values over Bayesian posteriors, we extract a mathematically exact feature attribution map from our optimal deterministic router (**CatBoost**). We employ cooperative Game Theory, specifically **Shapley Additive exPlanations (SHAP)**, to trace the global routing logic.

## 7.1 Mathematical Formulation of Shapley Values

In our early-stage predictor $g_{\boldsymbol{\theta}}(\boldsymbol{x})$, the 9 kinematic sensor axes represent players $N = \{1, 2, \ldots, n\}$ (where $n = 9$) in a cooperative game. The "payout" is the model's prediction output $v(N) = g_{\boldsymbol{\theta}}(\boldsymbol{x})$ for a specific temporal instance. The Shapley value $\phi_i$ quantifies the marginal contribution of the $i$-th sensor axis to the final prediction variance, averaged over all permutations of feature subsets $S \subseteq N \setminus \{i\}$. Mathematically, this allocation is defined as:

$$\phi_i(v) = \sum_{S \subseteq N \setminus \{i\}} \frac{|S|!(n - |S| - 1)!}{n!} \Big( v(S \cup \{i\}) - v(S) \Big) \tag{11}$$

The *local accuracy* property guarantees that the sum of Shapley values equals the difference between the model's output and the expected base value: $g_{\boldsymbol{\theta}}(\boldsymbol{x}) = \phi_0 + \sum_{i=1}^{n} \phi_i$. For tree-based ensembles

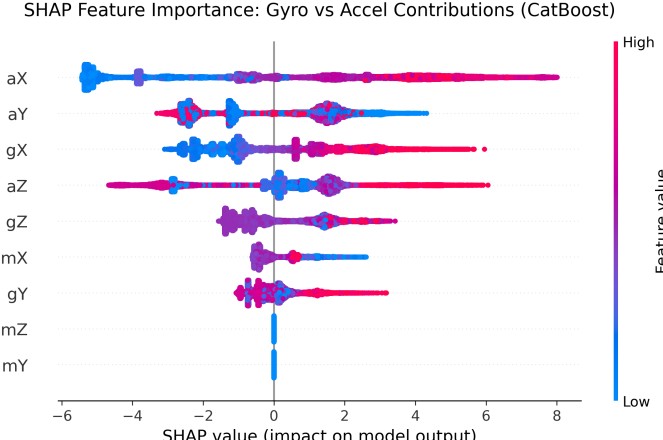

Figure 4: SHAP feature importance from the optimal router (CatBoost). Accelerometer axes (aX, aY, aZ) dominate over Gyroscope (gX, gY, gZ), instructing the Video Solver to prioritize translational stabilization (e.g., OIS) over rotational EIS.

like CatBoost, we utilize the optimized `TreeExplainer`, computing exact Shapley values in polynomial time $O(TLD^2)$, where $T$ is the number of trees, $L$ is the maximum leaves, and $D$ is the maximum depth.

## 7.2 Physical Insights and Routing Recommendations (OIS vs. EIS)

Figure 4 visualizes the global SHAP summary plot, where each dot represents the Shapley value of a sensor axis for a time window, colored by raw sensor magnitude. Analysis reveals that **translational kinematics strictly dominate rotational kinematics**: the top contributors are the accelerometer axes ($a_X, a_Y, a_Z$), with $a_X$ exhibiting significant marginal contribution spread. Conversely, gyroscope axes ($g_X, g_Z, g_Y$) possess a narrower impact, while magnetometer axes ($m_X, m_Y, m_Z$) show near-zero Shapley values, which is physically consistent as Earth's magnetic field remains invariant during high-frequency vibrations.

This physical insight directly informs the routing logic. While standard smartphone stabilization relies on **Electronic Image Stabilization (EIS)** to compensate primarily for *rotational* movements, our SHAP analysis confirms that severe mechanical anomalies (e.g., scooter vibrations or rapid walking) are heavily *translational*. Therefore, when the IMU predictor triggers the Heavy Video Solver ($a_t = 1$ from Eq. 2), the downstream solver must prioritize translational compensation mechanisms—such as **Optical Image Stabilization (OIS)** lens actuation, aggressive perspective cropping, or dense optical flow fields—over standard EIS. This demonstrates that game-theoretic XAI serves not merely as a diagnostic tool, but as a functional component of our cascaded routing architecture.

## 8 Conclusion and Future Work

Deploying heavy computer vision algorithms on mobile edge devices requires a shift towards conditional computation. We introduced the mathematical foundations of a cascaded inference paradigm for adaptive video stabilization. Utilizing a high-frequency, low-power IMU stream, we established a **cascaded router** bypassing heavy video solvers during stable kinematics, conserving computational and energy resources.

Our robustness benchmark yielded two pivotal conclusions. First, we identified a divergence between amortized batch throughput and true sequential latency. While LightGBM excels in batch processing, its API overhead is suboptimal for single-sample streaming. Thus, **CatBoost** proved superior, balancing low latency (0.44 ms) and accuracy.

Second, evaluating the **Bayesian 1D-CNN** (0.38 ms) exposed a vulnerability: under severe noise, the model suffers confident misclassification, contradicting the assumption that epistemic uncer-

tainty scales with OOD corruption. This proves singular reliance on Bayesian routing is hazardous, justifying our architecture pairing high-speed routers with orthogonal Isolation Forest monitors. Furthermore, game-theoretic SHAP analysis mapped anomalies to translational accelerometer shifts, optimizing the solver to prioritize OIS over EIS.

**Future Work:** Having established router viability, future work involves end-to-end integration with a deep learning-based video stabilization solver. We aim to deploy the Bayesian 1D-CNN on mobile NPUs to evaluate power consumption. Additionally, we will investigate dynamic calibration of the uncertainty threshold ($\epsilon$) via reinforcement learning, balancing visual smoothness and battery preservation based on the edge environment.

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
