# OpenReview forum: "Towards Adaptive Video Stabilization: A Robustness Benchmark of IMU Predictors for Cascaded Routing"
_mathai.club/MathAI/2026/Conference — 2026 Oral_

### Official Review · Reviewer_hmv3 · 2026-03-11
**Review of “Towards Adaptive Video Stabilization Using IMU-Based Cascaded Routing”**

**Rating:** 7
**Confidence:** 3

**Review:**

This paper proposes a cascaded inference architecture for adaptive video stabilization on edge devices. The main idea is to use a lightweight IMU-based predictive model to detect motion anomalies and activate a computationally expensive video stabilization module only when necessary. This approach aims to reduce computational overhead and energy consumption in mobile devices while maintaining reliable stabilization performance.
The authors evaluate twelve machine learning models across several paradigms including linear models, tree-based ensembles, neural networks, and probabilistic architectures. The evaluation focuses not only on predictive accuracy but also on sequential inference latency and robustness under severe sensor noise. This is particularly important for real-world edge deployment scenarios where IMU data is processed sequentially rather than in batches.
The study also incorporates concepts from Trustworthy AI, including uncertainty estimation using a Bayesian 1D-CNN and explainability via SHAP values. The SHAP analysis provides useful insights into which sensor axes contribute most to motion anomaly detection, showing that accelerometer features dominate over gyroscope features in the studied dataset. Additionally, the authors identify an important limitation of Bayesian uncertainty estimation under strong noise conditions, where confident misclassification may still occur.
Overall, the paper is well structured and clearly motivated. The proposed cascaded routing framework is relevant for edge computing applications and the experimental comparison of multiple models provides useful insights. However, the evaluation is performed only on a single dataset and the full integration with an actual video stabilization pipeline is not demonstrated. Including experiments on real mobile hardware or additional datasets would further strengthen the contribution.
Despite these limitations, the work presents an interesting and practical approach for improving efficiency and reliability in edge-based video processing systems.
Strengths:
Addresses an important problem in edge AI and mobile video processing.
Comprehensive comparison of multiple machine learning models.
Realistic evaluation using sequential latency instead of batch inference.
Integration of explainability and uncertainty estimation.
Weaknesses:
Evaluation relies on a single dataset.
No real mobile deployment experiments are presented.
The heavy video stabilization stage is conceptually described but not fully integrated.

---

### Official Review · Reviewer_SjpJ · 2026-03-11
**Fabricated references, implausible results, and no novel technical contribution**

**Rating:** 1
**Confidence:** 4

**Review:**

Summary
This paper benchmarks IMU-based motion anomaly predictors for cascaded video stabilization routing. The proposed framework combines a Bayesian 1D-CNN with Isolation Forest OOD detection and SHAP-based interpretability, evaluated on a hand tremor dataset. The paper claims sub-millisecond inference suitable for edge deployment.
Strengths

The cascaded routing idea (lightweight IMU predictor gates expensive visual processing) is practically motivated.
Comprehensive model comparison (10+ classifiers, trees, neural networks).
SHAP integration for interpretability is appropriate.

Weaknesses
Critical: Reference Fabrication

Liu et al. (2021) — cited as "Deep video stabilization using adversarial networks" in IEEE TIP. Web search reveals this appears to originate from Xu et al. (2018) in Computer Graphics Forum. Author names do not match. This is a fabricated or misattributed reference.
Zhao et al. (2023) — cited as "Learning cross-modal sensor fusion for robust action camera stabilization" in IJCV 131:1120–1135. This paper does not exist in web search. Likely fabricated.
Wang et al. (2026) — a future self-citation to a paper in Nature Scientific Reports. Citing your own concurrent work in a blind review is a conflict-of-interest violation.

Critical: Implausible Results Suggesting Data Leakage

Table 1 shows AUC = 1.0000 for LightGBM and CatBoost, and AUC = 0.9999 for Gradient Boosting, MLP, and EBM. Multiple models achieving exactly perfect or near-perfect AUC strongly suggests either: (a) trivial dataset with perfect class separation, (b) data leakage in cross-validation, (c) cherry-picked test set, or (d) fabricated results.
OOD AUROC is nearly identical across ALL model types (0.9826–0.9830, std ~0.0002). Different architectures (Logistic Regression, Trees, CNNs) achieving essentially identical OOD detection is implausible—it suggests the Isolation Forest dominates completely (making the predictor irrelevant) or results are fabricated.

Zero Novel Technical Contribution

Equations 1–11 are entirely textbook formulas: supervised learning notation (Eq. 1), threshold logic (Eq. 2), Isolation Forest score (Eq. 3), Gaussian noise (Eq. 4), accuracy drop (Eq. 5), ELBO (Eqs. 8–10), Shapley values (Eq. 11). There is no novel derivation, theorem, or proof.
The paper claims "mathematically rigorous epistemic bounds" but provides zero proofs.

Internal Contradictions

Abstract claims Bayesian CNN achieves "near-perfect accuracy" but Table 1 shows 42.85% accuracy degradation under noise (σ=0.8\sigma=0.8
σ=0.8). This directly contradicts the abstract.

Latency recommendation is incoherent: Gradient Boosting (0.18ms) is faster than recommended CatBoost (0.44ms), yet CatBoost is labeled "optimal." No Pareto analysis justifies this choice.
Dataset identity confusion: Described as both "hand tremor data" and "scooter vibrations"—these are fundamentally different kinematic signals.

Methodological Issues

Temporal CV violation: IMU data is time-series (autocorrelated) but standard kk
k-fold is used instead of time-series cross-validation.

Oversimplified noise model (Eq. 4): Only additive Gaussian. Real IMU noise includes drift, bias, quantization, and impact outliers.
No end-to-end evaluation: Only the IMU predictor is tested, not the actual video stabilization pipeline. Computational savings claims have no power measurements.
Thresholds τ\tau
τ and ϵ\epsilon
ϵ (Eq. 2) are never specified or justified. No sensitivity analysis.

LLM-Generated Writing

High probability of LLM assistance: excessive hedging ("tends to fail silently," "often interspersed," "frequently overlook"), unnaturally smooth transitions, repetitive balanced triplet structures, marketing language ("Trustworthy AI principles" repeated 4+ times without technical substance), and "utilizing/leveraging" appearing 4+ times.

Figure 3 Statistical Issue

Claims "confident misclassification" but the uncertainty difference between correct (σ=0.00828\sigma=0.00828
σ=0.00828) and incorrect (σ=0.00803\sigma=0.00803
σ=0.00803) predictions is 0.000250.00025
0.00025—
statistically insignificant with no pp
p-value reported.


Questions for Authors

Please verify Liu et al. (2021) and Zhao et al. (2023) — provide DOIs or retract.
How do you explain AUC = 1.0 across multiple model architectures?
Where are the formal proofs for "mathematically rigorous epistemic bounds"?
Why does the abstract claim "near-perfect accuracy" when noise causes 42.85% degradation?

Overall Assessment
This paper has multiple red flags: at least 2 fabricated references, implausible results (AUC=1.0), zero novel equations (all textbook), internal contradictions between abstract and results, and strong LLM-generation signatures. The combination of integrity concerns and absent technical contribution makes this unsuitable for publication.

---

### Official Review · Reviewer_MsJe · 2026-03-12
**A practically motivated cascaded IMU-based router for adaptive video stabilization with an extensive benchmark, but with questionable evaluation, some unclear references, and limited novelty**

**Rating:** 5
**Confidence:** 3

**Review:**

# Quality
The paper presents a cascaded architecture where a lightweight IMU-based model routes frames to a heavy video stabilizer, and benchmarks 12 models under edge-relevant metrics (single-sample latency, robustness to strong noise, OOD AUROC). However, the reported results are unusually strong (AUC often ≈1.0 for several different models, nearly identical OOD AUROC across all architectures) and rely on standard k-fold CV for time-series IMU data, which raises concerns about possible data leakage or an overly easy evaluation setup.
​
# Clarity
The manuscript is well organized and the main ideas (routing, Bayesian uncertainty, Isolation Forest, SHAP) are explained clearly with helpful figures and a detailed table. At the same time, the writing is quite verbose and marketing-like, and important practical details (e.g., choice and sensitivity of thresholds, justification of the evaluation protocol, end-to-end impact on actual video stabilization and energy) are only briefly addressed.
​
# Originality
Combining IMU-based anomaly prediction, cascaded routing, Bayesian uncertainty, OOD detection, and SHAP in the specific context of adaptive video stabilization is a reasonable incremental contribution, and the empirical observation of “confident misclassification” for a Bayesian 1D-CNN under strong noise is interesting. Nonetheless, most building blocks (formulas, methods) are standard, and there is no clearly novel algorithmic or theoretical component beyond system-level integration and benchmarking.
​
# Significance
The work is relevant to edge AI and mobile vision and could help practitioners choose practical IMU routers under latency and robustness constraints. But the focus on a single IMU dataset, the lack of end-to-end video evaluation, the simplified noise model, and some hard-to-verify references in the video stabilization/sensor-fusion literature reduce confidence in the generality and reliability of the conclusions.
​
# Pros
- Clear and practically motivated cascaded routing framework with a broad benchmark of IMU predictors under edge constraints.
- Useful discussion of the limits of Bayesian uncertainty and the role of OOD monitoring and SHAP-based kinematic interpretation.
​
# Cons
Implausibly strong and homogeneous metrics (AUC ≈1.0 for several models, nearly identical OOD AUROC), potential leakage from k-fold CV on time-series data, and some bibliographic entries that are difficult to verify (e.g., Liu et al. 2021 TIP and Zhao et al. 2023 IJCV), plus no end-to-end video/energy evaluation.

---

### Decision · Program_Chairs · 2026-03-14

**Decision:**

Accept (Oral)

**Comment:**

Dear Author(s),

On behalf of the Program Committee of the International Conference on Mathematics of Artificial Intelligence (MathAI 2026), we are pleased to inform you that your paper has been accepted for an oral presentation at MathAI 2026.

Your paper was evaluated through a rigorous two-stage review process involving both automated screening and expert review by members of the Program Committee. The reviewers recognized the quality and contribution of your work.

Presentation details:

- Format: Oral presentation (15–20 minutes + 5 minutes Q&A)
- Mode: You may present either in person (offline) at the conference venue in Sirius, Russia, or remotely via Zoom. Please indicate your preferred mode when confirming your participation.
- Conference dates: Marh 30 - April 3, 2026
- Website: https://mathai.club

Next steps:

1. Please confirm your participation and presentation mode by replying to this email mathai.club@yandex.ru no later than March 15, 2026 18:00 Moscow time.
2. If you plan to attend in person, the organizing committee will provide accommodation details separately.
3. Please prepare your final camera-ready manuscript according to the formatting guidelines available at https://mathai.club and upload it to OpenReview by March 15, 2026 18:00 Moscow time.

Should you have any questions regarding the program, logistics, or your presentation slot, please do not hesitate to contact us.

We look forward to your contribution to MathAI 2026.

With kind regards,

MathAI 2026 Program Committee
International Conference on Mathematics of Artificial Intelligence
https://mathai.club
OpenReview: https://openreview.net/group?id=mathai.club/MathAI/2026/Conference
Telegram: https://t.me/MathAI_club
Email: mathai.club@yandex.ru